# Will Internet Market Newness Improve Performance? An Empirical Study on the Internet Market Innovation of Offline Retailers in China

**Cheng Lu** [1,2,*] **, Tongyu Gu** [1,2] **, Jie Chen** [3] **and Zunli Liu** [4]

1   College of Fashion and Design, Donghua University, Shanghai 200051, China; gracegu@dhu.edu.cn
2   Key Laboratory of Clothing Design & Technology, Donghua University, Ministry of Education, Shanghai 200051, China
3   Antai School of Economics and Management, Shanghai Jiaotong University, Shanghai 200030, China; jiechjie@163.com
4   School of Management, Shanghai University of Engineering Science, Shanghai 201620, China; liuzl2@126.com
*   Correspondence: lc@dhu.edu.cn

**Abstract:** With the increasing impact of the Internet, the transformation of traditional retailers through Internet innovation has become a pressing issue. Based on the theory of innovation and marketing capability, this study investigated the relationships among adaptive marketing capability (AMC), Internet market newness, and innovation performance. The mediation effect of compromise on these relationships was also investigated. Through data analysis of 205 questionnaires in China, the current study confirmed that interaction between AMC and Internet market newness significantly affected innovation performance, and Internet market newness positively affected performance only when AMC was high. The results also demonstrated that compromise mediated the interaction with performance. AMC moderated not only the direct effects on performance by Internet market newness, but the effects on compromise by Internet market newness.

**Keywords:** Internet market innovation; adaptive marketing capability; compromise; retail

## 1. Introduction

With the rapid development of the Internet market, the innovation and transformation of enterprises have been accelerated. Internet-based emerging markets pose a challenge to the status of traditional markets [1]. The digital economy, represented by the Internet, has brought subversive changes to the applicability of traditional economic models [2,3]. The theory of remodeling and value creation of business models in the Internet era has been hotly debated academically [4–6].

China's traditional retail industry has suffered a greater impact. On behalf of Suning Tesco, Wangfujing, Yintai, Tianhong, and other well-known Chinese brick-and-mortar retailers built their own shopping sites, which was an iconic event for offline retailers to launch online sales channels [6–8]. Driven by the digital economy, new business forms of the retail industry are becoming more and more diversified and flexible. The level of innovation in production and business models has been continuously improved. Physical retail enterprises have launched the "New Retail" stage of expanding online sales channels from purely offline stores to deep integration of online and offline logistics [7]. The business model of Chinese retail enterprises presents four characteristics of transformation: deep online, socialization and fragmentation, being unmanned, and being platform-oriented [9]. In the era of mobile Internet, the typical new retail business model in China is that enterprises and customers are directly connected through products or community platforms. Sharing and interacting with customers, the community, as a platform constantly transmits emotional information and exchange resources and generates many connection dividends [10]. Chinese retail enterprises are building a new retail business model based on the Internet

platform that can double the enterprise value [11]. Yonghui Super Species, a new retail brand with online and offline chains in China, was rated as the top of the "Digital Case List of Retail Industry in 2018" by Harvard Business Review. Based on the Internet, the company reconstructed the business model of "people–products–places" which doubled the value of customers and enterprise. The floor efficiency of some mature stores of Super Species reached 150,000 yuan/year, which is 10 times that of traditional supermarkets (15,000 yuan/year), and the unit price of customers reached about 200 yuan [12]. However, these new business models have encountered some problems in the iterative process, such as market risk and the improvement of enterprise resource capabilities [9]. The Internet can improve the performance of retail enterprises, but the increasing complexity of management may also lead to the loss of profits for retail enterprises [8]. Metersbonwe, a Chinese clothing brand, has adopted many measures, such as O2O, online retailing, omnichannel retailing, intelligent stores, and social media. Unfortunately, its performance is still declining, and it has had to close stores. Under the background of the digital economy, can Internet market innovation really be profitable? What capabilities do retail enterprises need to improve to enhance their innovation performance?

## 2. Literature Review and Hypotheses

### 2.1. Internet Market-Based Innovation

Innovation refers to the introduction of new technologies, products, services, marketing concepts, systems, and management methods to stimulate a company's economic performance [13]. Innovation is generally considered as an important driving force for the success of enterprises [14,15]. Innovation is of great significance for sustainable development of enterprises [15]. In retailing, innovation has awakened considerable academic and business interest in recent years [13].

Innovation can be technique-based (e.g., when a product involves new technologies, new engineering design capabilities, and/or new production processes) or market-based (e.g., when the target market of a product requires a new service or sales infrastructure) [16–19]. In the retail industry, the literature has shown market innovation and technological innovation as two research routes [13]. However, compared with research on technological innovation, market-based innovation research is still in an early stage [15].

Market-based innovation refers to the creation of new customer value for emerging markets by taking innovative new approaches to any elements of the market offering [17,20]. Internet market-based innovation means that offline retail enterprises leave the mainstream market of the original service and create service value for the emerging Internet market by adopting different Internet marketing methods or Internet technology. In China, the traditional retailing model has been greatly challenged by rapid growth of the e-commerce sector [21]. In such a rapidly changing environment, developing digital marketing capabilities is highly important for companies to maintain competitiveness [22]. Therefore, there is strong urgency for Chinese retail enterprises to digitize marketing capabilities, formulate digital marketing strategies, and implement new retailing models [15].

In China, Internet market-based innovation is characterized by the concept of New Retail, which includes new retail philosophy, new retail format, and new retail theories. It is a combination of online and offline retailing supported by modern logistics [23]. In March 2017, the concept of New Retail was first raised in "New Retail in C Era" by Ali Research Institute, which defined it as a new retail format with consumer experience as the center and data-driven panretail. Its basic characteristics include three aspects: taking newness as the foundation, focusing on consumer demand, and reconstructing the relationship among people, products, and place.

Many retail enterprises are carrying out market innovation based on the Internet, changing the old business model, subdividing target customers and markets, and reconstructing consumption relations, supply chain systems, and consumption scenarios [23,24]. Retailers can also observe the Internet as a crowdfunding platform, search for products and services, and attract new customers [25] Retail enterprises also use technologies such

as big data analysis, cloud computing, artificial intelligence, and virtual reality to promote offline retailing and realize O2O business [26]. Internet market-based innovation of retail enterprises can also focus on reducing costs through different retail channels [26,27]. Moreover, the international market, such as international e-commerce, is a kind of market-based innovation of offline retail enterprises entering the Internet market [28].

Internet market-based innovation brings not only development opportunities but risks. On the one hand, Internet market-based innovations expand the operation scope of enterprises through acquisition and utilization of Internet resources [29]. It guarantees the sustainability of product demand, makes full use of the existing resources of enterprises, and expands the Internet market demand of established products. Moreover, it broadens the development space of enterprises as well as their access to resources, such as unique knowledge close to the Internet market. It provides knowledge, resources, and inspiration to promote the innovation ability of the Internet and the company's performance [30]. International development (e.g., cross-border e-commerce) is one of the innovative behaviors of enterprises entering the Internet market [28,31]. By effectively connecting with the external market, enterprises can enhance their Internet market capabilities, expand their possible Internet innovation behaviors, and thus improve their innovation performance [32]. Take Uniqlo, for example. Although it was a traditional clothing retail enterprise relying on offline chain stores, it was one of the earliest enterprises in China to cooperate with Tmall, an Internet retail platform, to explore the new Internet market. Through social networking services, it successfully expanded its market scope in China and became a successful retail enterprise with Internet market-based innovation [33]. Jollychic, a cross-border e-commerce enterprise, went further in Internet market-based innovation. It gained access to the United States, the Middle East, Europe, and some other markets through an Internet platform [28].

On the other hand, Internet market-based innovation brings risks, since consumers do not exist yet [17]. It is uncertain whether consumers in the new market would accept an enterprise's products. However, enterprises need to invest a lot of resources into the new Internet market [34]. Not all Internet market-based innovations are successful [8]. Tian, Yang, and Tian (2021) analyzed microdata at the retail enterprise level from 2005 to 2018. The results showed that during the process of offline retailing shifting to online retailing and the expansion of online retailing channels of retail enterprises, the profit loss is significant because of the substantial increase in operating costs and expenses [8]. Chang's (2020) research based on retail crowdfunding activities also came to a similar result, that is, that crowdfunding might have a negative impact on the retail industry, because the event owners may reduce their own revenue potential through running a crowdfunding campaign [35]. Like in all other breakthrough innovations, risks and opportunities coexist in Internet market-based innovation [8,26,34].

Some studies have shown that innovation may be negatively correlated with firm performance [8,36]. The latest literature has shown that compared with technology newness, market newness has a stronger positive impact on the product performance of new enterprises [37,38]. This brings both opportunities and challenges to this study. At present, it is unclear which factors predict market-based innovation and how market newness improves innovation performance.

### 2.2. Adaptive Marketing Capability

Adaptive marketing capability (shortened as "AMC" below) is the preventive, rather than responsive, marketing capability of enterprises to adjust strategies actively, and it is outside-in, initiative, and expansible [39,40]. AMC consists of three parts: (1) vigilant market learning that enhances deep market insights with an advance warning system to anticipate market changes and unmet needs; (2) adaptive market experimentation that continuously learns from experiments; and (3) open marketing that forges relationships with those at the forefront of new media and social networking technologies and mobilizes the skills of current partners [39,41].

Recent studies have shown that AMC is associated with competitive advantage of enterprises [42] in that it helps to improve the detection of potential risks and market opportunities [41] and indirectly affects the performance of enterprises [43]. AMC responds to the market faster than dynamic capabilities [41,44]. It enables a company to find and interpret key signals and take actions faster than its competitors in the business environment, thus gaining significant competitive advantage [45] and having an influence on international marketing performance [46]. Enterprises that adopted adaptive marketing strategies quickly and efficiently at the right time have become market leaders [39]. At the same time, the development of adaptability is often accompanied by the evolution of organizational form.

To traditional offline retail enterprises, the main bottleneck in the transformation is the huge gap between the marketing capability of traditional enterprises and the ability needed by the rapidly developing Internet market [21,47]. In the past two decades, the opening of markets has brought about competition between new and old entrants to the global market, because the barriers to entering some previously closed markets have been broken [48]. In the turbulent and rapidly changing market environment, AMC is the basis for enterprise transformation and upgrading [39], as well as the most important marketing capability to fill the gap between market change and enterprise capability [40,48]. In a high-competitive-intensity environment, AMC is a key success factor in promoting business model innovation [42], and AMC showed positive effects in relation to performance [48]. Therefore, AMC is conducive to the innovation and transformation of traditional retail enterprises.

AMC is helpful for seizing opportunities and avoiding risks. The opportunities for Internet marketization innovation lie in expanding the business scope of enterprises [29], entering new markets [28,31], and creating new consumption scenarios [33]. AMC encourages companies to have resources that are oriented to the requirements of the international environment through the search for new business opportunities, new markets, adaptation, and creation of products [49]. AMC includes the capabilityof adaptive market experiment, i.e., experimenting in the market and continuing to learn from these experiments [39]. Whether through the accumulation of knowledge or the development of the market itself, it is helpful to create new consumption scenarios, understand the new market faster, and meet the needs of new markets faster and better [48]. AMC also encourages the development of open partnerships and innovative resources, such as new technologies acquired through partners [39]. In terms of improving organizational competence, AMC obtains better organizational learning ability through open information exchange, thus transforming valuable external knowledge into internal knowledge and improving organizational innovation performance [38]. In a highly competitive market, AMC is essential, as it can enhance the value of consumer and maintain or increase their loyalty to the brand, thereby enhancing organizational performance [50]. Therefore, AMC can help enterprises to explore potential consumers more sensitively and adjust according to the market more quickly, so as to promote market innovation into better performance. AMC is very important, but the role and path of AMC in enterprise transformation have yet to be confirmed.

The risks of Internet market innovation lie in the uncertainty of consumers [17], the uncertainty of new market acceptance [8], fierce competition [26], and the cost increase caused by new investment [8,34,35]. AMC allows an organization to be attentive to the market, anticipating possible opportunities, flexing its strategies, and adapting proactively to the future development of the market, resulting in a performance superior to its competitors [41]. In the uncertain new market, enterprises with AMC do not invest directly, but utilize external resources and capabilities to realize integration and cooperation between integrated network and internal resources [39,48]. It can effectively reduce the cost increase and risk caused by Internet market innovation [48]. Therefore, adaptive marketing capability can not only reduce costs through marketing experiments and open cooperation, but control the risks in market innovation, thereby reducing losses. Because of this, the following hypothesis was elaborated.

**Hypothesis 1 (H1).** *AMC plays a positive moderating role in the impact of Internet market newness on project performance. That is to say, under the condition of strong AMC, project performance is better for strong Internet market newness than for weak Internet market newness; under the condition of weak AMC, the impact of Internet market newness on performance is reduced.*

*2.3. Compromise*

Compromise is a common method of organizational politics [51,52]. Compromising is a strategic approach based on the need to reach a consensus among several key institutional participants with different interests and perspectives [53]. Concretely, compromise refers mainly to the necessary and reasonable transfer of interests by all parties concerned through negotiation to seek a new and acceptable balance. It can prevent the intensification of existing conflicts, achieve win–win cooperation among all parties, and ensure continuous cooperation [52].

Compromise is inevitable in the process of organizational change [54,55]. In general, all aspects of change, whether positive or negative, are often associated with organizational politics, especially when the change is associated with the reorganization of resources and power in the organizational structure [56,57]. Compromise sometimes helps organizations to relieve conflict and allow new projects to continue [58]. In addition to environment of organizational change, compromise strategy is also influenced by organizational stakeholders [58,59], social culture [60], leaders [61–63], etc. Organizational politics is ubiquitous in enterprises, and it is more intense in the process of innovation and transformation. However, in the literature on enterprise innovation and transformation, there has been little research on organizational politics.

The existing literature has confirmed that compromise is of great value in the process of innovation [64]. Compromise is considered one of the main strategies that can be used to manage tension [55]. Compromise is to resolve conflicts and promote the implementation of projects [58,65]. Research on China has shown that compromise is an important way for Chinese managers to deal with conflicts [66] and an effective way to resolve contradictions at the present stage of society [52].

When Internet market-based innovation projects are brand new to companies, different stakeholders can have different perspectives on them [58]. These projects often have a tremendous destructive effect on senior managers' control over existing business and resources, thus threatening their power base [67]. For example, projects targeting the new Internet market may require new sales processes and new service infrastructure, which in turn require a different power structure than the existing one in the company. If new sales processes and new service facilities are under the control of new team leaders, then the newness of the Internet market means that existing market managers may face loss of power and resources [54]. This threat leads to attempts to resist innovation [67].

When an innovation project team realizes that the project is facing resistance, it can take certain political measures to ensure the implementation of the project. The most basic political action is "compromise" [67]. Compromise in the implementation of new projects in the Internet market can take the form of adjustment of the existing market rather than pursuit of brand-new markets (for example, trying to achieve balanced development of the online and offline retail market instead of pursuing a single Internet retail market).

When the compromise strategy is adopted in the implementation of innovation projects, it is mainly to ease the organization–stakeholder relationship [58]. Companies with AMC can reduce internal conflicts through more flexible and open cooperation [39,41]. For example, conflicts can be reduced by looking for information providers rather than cooperative developers [68]. Negotiating with senior managers and making concessions on parts of innovation projects can lead to approval of innovation projects [56].

At the same time, breakthrough market-based innovation often has greater risks [17]. AMC also means better organizational learning ability through open information exchange, thus transforming valuable external knowledge into internal knowledge. Learning can help stakeholders to understand the new Internet market and make it easier for them to make

concessions to their inherent positions, thus improving compromise. AMC drives managers to take full consideration of the resistance and conflict within enterprises while pursuing innovation. It makes managers center around customers, establish information communication between old and new departments, and conduct marketing experiments [39]. The unreasonable and ineffective part of newness with insufficient resource support is abandoned, thereby reducing Internet market newness and increasing the compromise. Because of this, the following hypothesis was elaborated.

**Hypothesis 2 (H2).** *AMC plays a positive moderating role in the impact of Internet market newness on compromise. That is to say, under the condition of strong AMC, compromise is higher for strong Internet market newness than for weak Internet market newness; while under the condition of weak AMC, the impact of Internet market newness on compromise is reduced.*

Traditional retail enterprises have to go through a series of struggles and compromises before reaching a balance between the old and new retail models [69]. Empirical evidence, in general, has shown that conflict undermines new product quality, disrupts delivery schedules, increases development costs, and leads to relationship termination [68,70,71]. Appropriate compromise to meet the requirements of the management team is an effective way to implement Internet projects, although this may be limited because of the risk of losing long-term, reliable support from stakeholders who want to see their interests or expectations more fully achieved [55,72,73].

Compromise can help overcome obstacles and facilitate the innovation process [56]. Aiming at adapting to the new market, adaptive marketing capabilities enable Internet project teams to improve as much as possible within managers' control. Appropriate compromise to meet the requirements of a management team is an effective way to implement Internet projects. Compromise under the influence of adaptive marketing capability is also in line with the needs of enterprise development, which helps to achieve project performance. Because of this, the following hypothesis was elaborated.

**Hypothesis 3 (H3).** *Compromise plays a mediating role in the impact of the interaction between AMC and Internet market newness on performance.*

This study preliminarily drafted the conceptual model shown in Figure 1 based on the foregoing theoretical review and hypotheses.

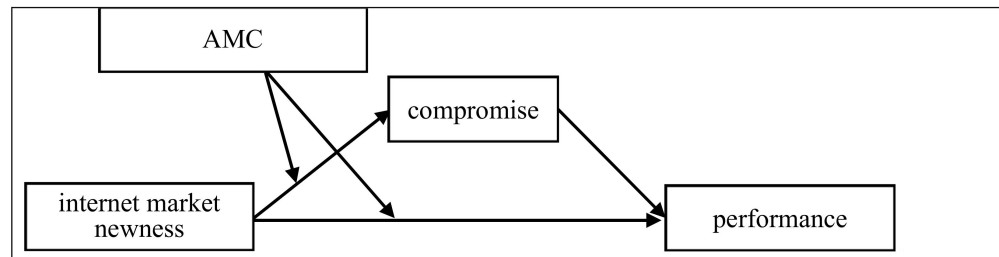

**Figure 1.** Conceptual framework.

### 3. Research Methods

*3.1. Variables and Measures*

To ensure the reliability and validity of the questionnaire, all items used for measuring this study's variables were selected from mature studies in the literature. All the questions were translated into Chinese. Professional translation and retranslation were used to ensure that all scales being used were consistent in meaning with the original scales. Ten managers participated in the pilot test. Based on their feedback, a small number of measurement items were revised to improve the clarity of the measurement. A seven-point Likert scale was adopted for all questionnaire items, as shown in Appendix A.

Internet market newness. The items of Sethi, Iqbal, and Sethi (2012) [54] and Danneels and Kleinschmidt (2001) were used to inquire about the differences between Internet innovation projects and other business departments [18]. The item scale mainly evaluated the newness of Internet innovation projects in terms of target market and required sales system and customer service infrastructure compared with those of other projects.

AMC. This measurement was based on Day (2011) [39] and Polat (2015) [74]. The scale comprised 19 items grouped into three dimensions: vigilant market learning, adaptive marketing experiment, and open marketing.

Compromise. It was measured using a 3-item scale adapted from Sethi, Iqbal, and Sethi's (2012) scale [54]. This part examined the compromise actions taken to ensure project approval in the process of Internet development of enterprise products or services.

Performance. The performance was measured using a scale adapted from Zhou, Yim, and Tse (2005) [19]. The scale comprised 4 items. The research was not to inquire about the overall performance of companies, but to inquire about the performance of Internet projects launched by offline retail enterprises. The respondents were asked to assess the product quality and customer value of their companies' innovative products or services compared with those of competitors' projects. A comparative scale was used so that the appraisal of performance would not be affected by specific product categories or industries.

Moreover, the respondents were asked about basic information on their retail enterprises, including the sizes and histories of the retail enterprises and the lengths of the respondents' employment.

### 3.2. Data Collection and Sample

The research scope was retail enterprises in China, including local enterprises and foreign-funded enterprises. Among the top 100 retail chain enterprises in China and the top 500 enterprises in the world, enterprises were selected according to the standards of "operating offline retail in China" and "launching Internet innovation projects in China". The survey was carried out through three channels. First, the questionnaires were distributed through connections in domestic retail industry to major retail brands; second, the questionnaires were sent to retail enterprises through member institutions of the Shanghai Commerce Commission; and third, we commissioned research companies to investigate some retail enterprises. To ensure that all the respondents were in charge of or involved in Internet innovation projects, respondents had to fall within one of the three following categories: (1) those who participated in the development of Internet projects in retail industry in the previous two years; (2) professionals who participated in Internet projects; (3) managers who were involved with Internet projects. The survey was conducted simultaneously through a paper questionnaire or an e-mail questionnaire, depending on the work habits of the respondents.

A survey was conducted about the Internet transformation of traditional offline retail enterprises in China. Following previous researchers' data collection methods for innovation research [19], the model hypothesis was validated at the level of the strategic business unit (SBU) in the retail product category. To ensure the validity of the questionnaire, for each company sample, we called the company to identify a senior manager (e.g., marketing director, general manager, and regional brand manager) as the key information provider [19]. The electronic questionnaire and cardboard questionnaire were distributed to key information providers and then collected after one week. The key information providers further screened the research respondents to ensure that their abilities and knowledge about the company matched the survey contents. Each respondent independently completed a company survey report. Finally, the questionnaire was returned anonymously.

A total of 211 questionnaires were collected, of which 205 were valid. The invalid questionnaires were mainly those returned by respondents who filled in the same value continuously or failed to pass the screening test. Retail product categories included clothing, bags, accessories, food and beverages, daily use, automobiles, digital, science and



technology, life, culture, commercial real estate, children's goods, and comprehensive categories. Local retail enterprises in China accounted for 65.1% of the sample, and foreign retail enterprises in China accounted for 34.9%. Companies engaged in regional retail business accounted for 40.2%, and those engaged in global retail business accounted for 57.9%.

Retail enterprises in the survey had different forms of Internet business, with "the establishment of an independent Internet business department" (43.1%) and "the starting of Internet business in an existing department" (26.8%) as the main forms, accounting together for 69.9% of the Internet transformation of the offline retail industry. "Establishing a new independent Internet business company" (7.7%), "entrusting third-party companies to operate Internet business on their behalf" (11.6%), "joint venture with professional Internet technology company to form a new company" (6.2%), and "other" (4.6%) were the other ways to carry out Internet innovation projects.

Respondents had different job backgrounds, including project researchers and developers (41.1%), project professionals (20.6%), project managers (14.8%), marketing department managers (14.8%), and senior managers (8.6%). The respondents held either mid-level or senior positions in retail companies. They had an average of 7 years' work experience. The different backgrounds of the subjects are not discussed, since they held no significant differences in terms of the variables of the study.

## 4. Data Analysis

### 4.1. Correlation Analysis, Reliability and Validity Analysis

A confirmatory factor analysis with maximum likelihood estimation was conducted using AMOS to examine the reliability and validity of the latent variables, including performance, compromise, Internet market newness, and AMC. The measurement model fit was good ($\chi^2/df$ = 1.962; CFI = 0.856; RMSEA = 0.068), and all standardized factor loadings were greater than 0.50, which is the general baseline ($p < 0.001$) [75]. All composite reliabilities and Cronbach's alpha scores were greater than 0.8, and average variance extracted (AVE) measures exceeded 0.65 (acceptable, though not ideal), offering evidence of convergent validity [76]. The AVE for each factor exceeded any of the squared pairwise correlations involving the factor, confirming the discriminant validity of the measures [77].

The correlation coefficient between AMC and Internet market newness was 0.190, the correlation coefficient between AMC and compromise was 0.127, and the correlation coefficients between Internet market newness and compromise was 0.364. The mean, standard deviation, AVE value, composite reliability, Cronbach's alpha value, and correlation coefficient of each variable is given in Table 1.

**Table 1.** Variable correlation coefficients and descriptive analysis.

| Measurement Construct | 1 | 2 | 3 | 4 | 5 | 6 | 7 |
|---|---|---|---|---|---|---|---|
| 1. performance | 0.880 | | | | | | |
| 2. compromise | 0.224 *** | 0.819 | | | | | |
| 3. Internet market newness | 0.186 ** | 0.364 *** | 0.807 | | | | |
| 4. AMC | 0.254 *** | 0.127 † | 0.190 ** | 0.839 | | | |
| 5. size of company | 0.246 *** | −0.129 † | −0.172 ** | −0.057 | | | |
| 6. history of company | 0.075 | −0.091 | −0.193 ** | −0.193 ** | 0.586 *** | — | |
| 7. no. of working years | 0.109 | 0.003 | −0.041 | 0.006 | −0.007 | −0.029 | — |
| mean value | 5.507 | 4.538 | 3.968 | 4.899 | 3.97 | 4.32 | 1.31 |
| standard deviation | 1.020 | 1.090 | 1.302 | 1.476 | 2.107 | 2.368 | 0.624 |
| AVE | 0.774 | 0.670 | 0.651 | 0.705 | — | — | — |
| composite reliability CR | 0.872 | 0.859 | 0.862 | 0.877 | — | — | — |
| Cronbach's alpha value | 0.867 | 0.822 | 0.887 | 0.872 | — | — | — |

Note: † $p < 0.1$; ** $p < 0.01$; *** $p < 0.001$. The number on the diagonal line is the square root of the average variance extracted (AVE) of the measurement construct.

### 4.2. Regression Analysis

To test the hypothesis of the moderating effect of AMC on the relationship between Internet market newness and performance, the data were analyzed using regression. Performance was taken as the dependent variable. AMC and Internet market newness were taken as independent variables and centralized. The results showed that AMC affected performance ($\beta = 0.213$, $t = 4.383$, $p = 0.000 < 0.001$); the impact of Internet market newness on performance was not significant ($\beta = 0.030$, $t = 0.578$, $p = 0.564 > 0.1$); and more importantly, there was a significant interaction between AMC and Internet market newness ($\beta = 0.091$, $t = 2.553$, $p = 0.011 < 0.05$) (Table 2).

**Table 2.** Regression results of Internet market newness's and AMC's influences on project performance.

| Variable | Moderating Effect of AMC | | | |
|---|---|---|---|---|
| | Performance | Performance | Compromise | Performance |
| Intercept | 4.884 *** | 4.728 *** | 4.517 *** | 3.778 *** |
| Independent variable | | | | |
| Internet market newness | | 0.030 | 0.285 *** | −0.030 |
| AMC | | 0.213 *** | 0.088 † | 0.195 *** |
| Internet market newness × AMC | | 0.091* | 0.085 * | 0.073 * |
| Mediation variable | | | | |
| Compromise | | | | 0.2104 ** |
| Control variable | | | | |
| Size of company | 0.148 *** | 0.124 ** | −0.054 | 0.136 *** |
| History of company | −0.044 | −0.000 | 0.029 | −0.006 |
| No. of working years | 0.176 | 0.202 † | 0.056 | 0.190 † |
| Test result | | | | |
| $R^2$ | 0.080 *** | 0.169 *** | 0.160 *** | 0.212 *** |

Note: † $p < 0.1$; * $p < 0.05$; ** $p < 0.01$; *** $p < 0.001$.

To analyze the relationship between AMC and Internet market newness, the interaction was further decomposed. When the AMC was strong (one standard deviation higher than the average), the performance of enterprises with high Internet market newness was significantly higher than that of those with low Internet market newness ($\beta = 0.165$, $t = 2.226$, $p = 0.027 < 0.05$); when the AMC was weak (one standard deviation lower than the average), there was no significant difference in performance between enterprises with high Internet market newness and those with low Internet market newness ($\beta = -0.104$, $t = -1.397$, $p = 0.164 > 0.1$). The results of each group are shown in Figure 2. H1 was supported.

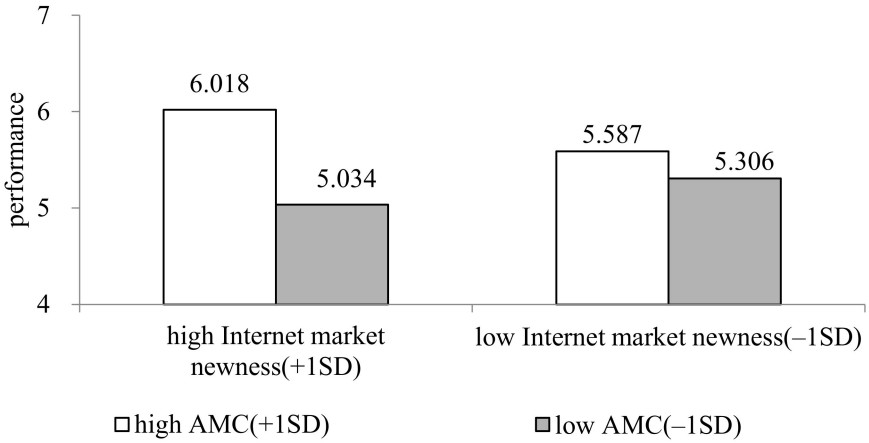

**Figure 2.** Impact of interaction between AMC and Internet market newness on performance.

To analyze the moderating role of AMC in the impact of Internet market newness on compromise, regression analysis was used, with centralized AMC and Internet market newness as independent variables and compromise as the dependent variable. The analysis results are shown in Table 2. Internet market newness had a significant impact on compromise ($\beta = 0.285$, t = 5.032, $p = 0.000 < 0.001$). AMC had a marginally significant impact on compromise ($\beta = -0.088$, t = 1.672, $p = 0.096 < 0.1$). There was significant interaction between AMC and Internet market newness ($\beta = 0.085$, t = 2.209, $p = 0.028 < 0.05$). To test hypothesis H2, the interaction was further decomposed. At high AMC levels (one standard deviation higher than the average), compromise was higher when Internet market newness was high than when it was low ($\beta = 0.411$, t = 5.134, $p = 0 < 0.001$). However, at low AMC levels (one standard deviation lower than the average), compromise was marginally higher when Internet market newness was high than when it was low ($\beta = 0.159$, t = 1.976, $p = 0.050 < 0.1$). We concluded that under the condition of high AMC, as Internet market newness increased, compromise was improved. H2 was supported. The results are shown in Figure 3.

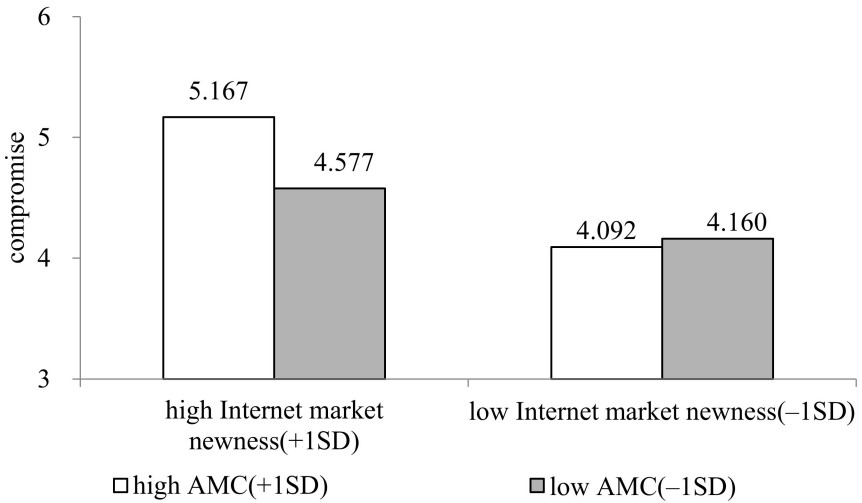

**Figure 3.** Impact of interaction between AMC and Internet market newness on compromise.

Next, H3 was tested, that is, whether interaction between AMC and Internet market newness affected performance through compromise. Conditional process analysis (Hayes, 2013, Model 7) was used, with AMC, Internet market newness, and Internet market newness × AMC as independent variables, compromise as a mediating variable, and performance as the dependent variable. The bootstrapping method was used to test the mediating effect. The results showed that the indirect effect of compromise was marginally significant (the indirect effect of the highest order interaction was 0.018, 95% CI = (0.000, 0.054)), indicating that the interaction between AMC and Internet market newness affected compromise, and in turn, project performance. H2 was supported. The mediating effect of compromise under different Internet market newness conditions was specifically analyzed. It was found that under the condition of high Internet market newness, compromise mediated AMC and performance, with indirect effect as 0.087 (95% CI = (0.024, 0.182)); under the condition of low Internet market newness, the indirect effect of AMC on performance through compromise decreased to 0.034 (95% CI = (0.001, 0.092)). H3 was supported.

So far, we have discussed the impact of Internet market newness, AMC, and compromise on the performance of offline retail industries when they attempt to launch Internet market-based innovation projects. The mediating role of compromise and the moderating role of AMC were verified. Specifically, when a company had strong AMC, its performance with strong Internet market newness was better than with weak Internet market newness. It was intraorganizational compromise guided by strong AMC that helped to improve

innovation performance. However, when a company lacked AMC, the impact of Internet market newness on performance disappeared. There was no significant difference between the performance of enterprises with high and low Internet market newness. In short, the theoretical framework shown in Figure 1 was verified. The interaction between AMC and Internet market newness significantly affected performance through compromise.

## 5. Conclusions and Discussion

### 5.1. Conclusions

Against the background of the era in which the Internet has a huge impact on traditional industries, this paper took the "going online" of the offline retail industry as the research object. From the perspective of facilitating the implementation of Internet market projects from within enterprises, it examined the impact of compromise and AMC on the process of the transformation from offline retail companies to online retail companies. The results showed as follows.

First, the results of this study affirmed the positive impact of Internet market innovation on performance and found the marginal conditions for innovation to performance. According to the theory of innovation, innovation has positive significance to enterprises [14,15]. Market innovation and technological innovation are two research routes of innovation [16–19]. In the era of the digital economy, how retail enterprises carry out Internet technological innovation has been widely discussed, while the research on how retail enterprises can upgrade their business through Internet market innovation has not yet matured. The present research found that AMC was the boundary condition of innovation performance. Specifically, there was no significant relationship between Internet market innovation and performance when AMC was low, while Internet market innovation significantly positively impacted performance when AMC was high.

Second, AMC is the core competitiveness of promoting the transformation of the retail industry. Unlike static and dynamic capabilities, AMC is a future-oriented capability put forward by [39]. AMC is responsible for an organization's active skills, facing the feelings and actions of the signals transmitted by the market, constantly learning from market experiences, coordinating internal resources to quickly respond to these changes [39], and taking positive actions on the organization's performance when the environment is more competitive [44]. In a turbulent environment, especially when dealing with a market with such complexity and transformation as the Internet market, AMC is very important to traditional retail enterprises [22,48]. This study revealed that AMC and Internet market innovation influenced performance via compromise. Specifically, Internet market newness could not improve innovation performance when AMC was low. When AMC was high, there was more information exchange, experimental innovation, and open cooperation between enterprises and the environment. Therefore, firms with high compromise had stronger responses and greater adjustment to Internet innovation projects. The new Internet market will lead to higher compromise and ultimately higher performance.

### 5.2. Discussion

Under the tremendous impact of the Internet, the innovation and transformation of the retail industry has become a topic of great concern [1]. However, most of the existing literature has focused on business model innovation [4–6], less on retail transformation from the perspective of enterprise marketing capabilities or on the relationship and boundary conditions between innovation and performance in the Internet market. In addition, previous studies have not considered the relationship among resource capability, organizational politics, and innovation performance. In short, this study contributed to the literature in the following aspects.

On the one hand, this study filled a gap in that innovation had different effects on performance in previous studies. The present findings are important to the development of marketing and innovation theory. Most previous research focused on the effects of dynamic capabilities in promoting innovation performance and overlooked the importance of AMC.

There has been a lack of research on the combined effect of AMC and innovation. Although innovation is considered to be the key driving force for enterprises to maintain long-term and lasting competitive advantages [78,79], some studies have shown that innovation may be negatively correlated with firm performance [8,36]. The theoretical framework of "capability–activity–performance" [44,80] suggests that the enhancement of an enterprise's AMC is a key mechanism for driving the achievement of superior performance. This study verified the significant promoting effects of innovation on performance and clarified AMC as the marginal condition (i.e., moderating effect). In this way, this research promoted the development of market innovation theory in China. It also shed a new light on the innovation–performance research stream and provided some new directions for future research on the relationship between an organization's AMC and organizational change.

On the other hand, this study analyzed the innovation process from the perspective of organizational behavior. The research expanded the understanding of organizational politics. Organizational political behavior in new project development is significant. It becomes particularly prominent when resource competition is involved in the process of new project review and approval [54,67]. The role of organizational politics in the transformation of enterprise was considered negative: innovative projects bring about threats, pressure, resistance, and other organizational political behavior, to cope with which innovators adopt compromise [54]. Organizational politics are often regarded as the implicit rule of enterprises, with certain negative effects [54]. On the contrary, this study found that although compromise is a kind of passive decision making, it played a positive role in the impact of Internet market newness and AMC on performance. This inference was different from the results of previous studies that mainly supported the negative effect of organizational politics on innovation.

*5.3. Managerial Implications*

The conclusions can be used as a guide for traditional retailers to implement Internet market innovation and transformation projects. The results also have several implications relevant to the management of enterprises. First, the effective development of AMC was shown to be the key to successful innovation and transformation in retail enterprises. To ensure profit from innovation, enterprises must give full play to AMC's role in improving the performance of market-innovation activities. In the process of transformation and upgrading of the retail industry, one should not blindly pursue the novelty of Internet projects but pay more attention to the matching of enterprise marketing capabilities and innovative projects. This study proposes that marketing capabilities such as AMCs should be permitted to drive the Internet market innovation of enterprises. In a highly competitive and intensive environment, AMC is a key success factor in promoting business model innovation [42] and sustainable development [48].

Second, the implementation of innovation projects and good performance also requires enterprises to make rational use of compromise to solve conflicts. Compromise is an effective way to deal with internal conflicts—not blind compromise, but a reasonable compromise with AMC. When the Internet market newness of a project is so strong that it leads to internal resistance, managers and innovators can properly use compromise as an organizational political method to promote the implementation and performance of the project. This means that with strong AMC, a company's Internet market innovation project is strictly examined by the existing marketing team and management for power or control, so it is easier for the innovation project team to compromise to ensure that the innovation project can proceed. This is not a bad thing, because the relationship between innovative projects and the Internet environment is more closely examined. Therefore, compromise is helpful to obtain good performance.

*5.4. Limitations and Future Research*

There are some limitations in the current research. First, because the limitation of research sampling time, we studied only Internet market newness and the short-term

impact of AMC on performance and did not consider the long-term impact on brand reputation and competitiveness. In the future, we can conduct research related to the long-term impact of the retail industry's transformation. Secondly, we took only offline retail enterprises in the Chinese market as samples. To overcome these limitations, other studies can be done with the inclusion of other countries in the sample, allowing a comparison between the differences in the performance of AMC and market innovation performance. As seen in the whole study, many variables can be considered in future surveys, such as demographic and organizational management characteristics of companies, for example, distinguishing them by country. It is also possible to carry out sectoral research and research with longitudinal characteristics. Future study is needed to control these variables to support our research model more precisely. The researchers of this study hope that future studies can address these limitations.

**Author Contributions:** Conceptualization, C.L., J.C. and Z.L.; methodology, C.L. and J.C.; investigation, C.L., J.C. and Z.L.; data analysis, C.L. and Z.L.; writing—original draft preparation, C.L. and T.G.; writing—review and editing, C.L.; funding acquisition, C.L. and J.C. All authors have read and agreed to the published version of the manuscript.

**Funding:** This study was supported by the Fundamental Research Funds for the Central Universities (grant NOs. 2232021G-08, 2232021E-03), the National Natural Science Foundation grants of P.R. China (grant NOs. 71840009, 71832008), and key scientific and technological innovation plan of action of soft science projects by Science and Technology Commission of Shanghai Municipality (grant NO. 0692100800).

**Institutional Review Board Statement:** Not applicable.

**Informed Consent Statement:** Not applicable.

**Data Availability Statement:** The data presented in this study are available on request from the corresponding author.

**Conflicts of Interest:** The authors declare no conflict of interest.

## Appendix A

Measure Items
Internet market newness
The statements describe how different the initial new product was from the other products of the business unit (7 = "strongly agree", and 1 = "strongly disagree").

1.   Compared with our past market, the Internet market is brand new.
2.   The Internet market is new for our business unit.
3.   The Internet requires a new sales force system.
4.   The customer service infrastructure required for the Internet market was new.

Adaptive Market Capability

1.   We focus on the lives of current, prospective, and past customers.
2.   We observe customers' reactions to social networks and the social media space.
3.   We do not have any preconceived ideas of customers.
4.   We are open to the potential needs of our customers.
5.   We have an ability to sense weak signals from the periphery and take actions against them.
6.   We have an experimental mindset.
7.   We are willing to challenge existing beliefs.
8.   We collate and share ideas and successful practices across the organization.
9.   We are willing to learn from peers.
10.  We are willing to learn from the experience of pioneers in the field.
11.  We are looking for talented people with rich knowledge and wide knowledge.
12.  We are constantly acquiring new capabilities.
13.  Our company can get richer, more diverse, and more microscopic responses.

14. The extension of the periphery can bring new insights to the company.
15. We welcome our employees to bring some new ideas to the company.
16. Expertise can be obtained from numerous sources.
17. We are allowed to communicate and exchange market information with the outside of the company.
18. Professional knowledge can be acquired from various sources.
19. In reality, as a joint venture, open market cooperation can sometimes be disappointing.

Compromise.

The statements are related to the actions that were taken at the initial stages of the product development process to ensure the approval of the product for development (7 = "strongly agree", and 1 = "strongly disagree").

1. To be approved at gate reviews, the product definition had to be compromised to satisfy the demands of senior managers present at gate reviews.
2. To be approved at gate reviews, the product had to be significantly modified to cater to competing demands from various company stakeholders.
3. After being approved at gate reviews, the product parameters differed significantly from those prior to gate reviews.

Performance.

Our firm's overall performance compared with major competitors over the past year (1 = "far below the competitors", and 7 = "far above the competitors")

1. Sales growth in the past two years.
2. Return on investment.
3. Profit level.
4. Market share.

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
