# Peer review of "Will Internet Market Newness Improve Performance? An Empirical Study on the Internet Market Innovation of Offline Retailers in China"

_sustainability, doi:10.3390/su132212619_

Round 1

Reviewer 1 Report

This paper represents a relevant contribution to the effect of adaptive marketing capability and compromise in Internet market-based Innovation of offline retail companies in China. Context itself seems reasonable, and certainly opens an avenue for discussion. The introduction to the paper nicely suggests that there still are gaps in knowledge that can be filled with the type of research conducted in this study. However, in my opinion some further considerations are the following:

  • I suggest considering a general, integrative theoretical model to present a conceptual framework previously. In the introduction and particularly in the literature review sections, I suggest considering a general, integrative theoretical approach to present the conceptual framework previously, and then write the paper from the angle of the specific chosen approach.
  • I find it a little confusing to read in the Introduction Section some considerations regarding the three specific research questions that authors have used in this paper and, afterwards, to describe three hypotheses. I suggest that these considerations could be included according not in the Introduction, but in a literature review section. To eliminate redundancy, please write only research questions or hypotheses, not both.
  • It would be recommended to move table 1 to the next section: Results. in addition, I suggest renaming this section as follows: Results and Findings.
  • The last section is too brief and requires more elaboration in previous sections. Particularly, a Conclusion section regarding the limitations and ways in which this research with differing intervention contributes to managerial and theoretical implications in the study is required. To the end of the paper, I suggest including several subsections renaming this section as follows Conclusions, Limitations, and Implications.
  • In my opinion, results are more indicative rather than representative. More limitations and managerial contributions of the study in terms of the generalization of the findings should be added.
  • Please, update references in the paper.

Reviewer 2 Report

Dear Authors,

Thank you for submitting your paper to the "Sustainability". I read it with great interest. Unfortunately, I believe that the article needs a lot of work before it is submitted for re-evaluation. I noticed many major and minor flaws in it. The latter include, for example, editing and linguistic errors, the failure to label Sections 2.3 and 2.4, or the inconsistent spelling of the abbreviation AMC (once uppercase, once lowercase).

Among the more significant flaws, I would like to mention the structure of the study. It is chaotic, especially within the introduction and the literature review. The threads taken up suddenly stop, then wholly different ones appear, only to return at the end to the lines initially analysed (in 2.1 and 2.2.). The literature review is superficial, incomplete and lacks cause-effect relationships. There is also a lack of more recent publications (as of 2019) that point to contemporary AMC or Internet Market-based Innovation approaches (ImbI). It would be worthwhile to internationalise your study more strongly by drawing on more foreign sources. 

I would suggest a different division of content and structure of the text. In the "Introduction", I would read about the situation of the offline retail industry in your country (how do you define it, does it have a raison d'être, in general, today in 2021? Does it generally exist today?). Then I would like to read about how you link the offline retail industry with AMC, the IMbI trade-off (with a definition of the phenomena already here). I also think that the "Introduction" is not a good place to pose research questions. These should emerge from the literature review, matching the research gaps observed. However, you have not demonstrated any gaps at this stage, and the literature review itself was not sufficient.

In the theoretical part (chapter 2), I would suggest the following: presenting the innovation theory with a clear and quick transition to IMbI (its definitions, characteristics, classification and putting it in the context of offline retail companies). I would then focus on AMC theory (definitions, features types and framing it in the context of IMbI, including IMbI in offline retail companies). Then I would refer to the compromise (definitions, classification features and preparing it in the context of IMbI, including IMbI in offline retail companies). With this, I would diagnose academic gaps and formulate hypotheses and research questions.

My doubts connect with the description of the methodology used. For the clarity of the paper, please present the whole research process precisely: 

- construction of the questionnaire (when you did it, who did it, the rationale for the selection of tools, final form)

- testing of the questionnaire (when tested, how tested, how you selected the experts, what you changed after their opinion)

- the final version of the questionnaire

- the questionnaire's distribution (selection rules, selection criteria, distribution time, return rate of completed questionnaires). 

I also would like to ask about the ethical aspects of your work. Did you anonymise the questionnaire? Was it approved by the appropriate committee, etc.? 

It is unclear why you distributed the questionnaire abroad. What use was made of data from, e.g. New York, Florence, if the basis of your analysis is China? 

After all, you do not highlight any differences between your country and the rest of the world anyway. 

In the "Discussion", you do not precisely answer the research questions posed earlier.

I also noticed some more limits to your text, e.g. unrepresentativeness, dispersion of the research sample.

Considering the scale of the necessary changes, and above all the methodological inaccuracies, the fragmentary literature review and the not very up-to-date bibliographical data, I am very sorry to say that it is too early to publish your study.

I hope, however, that you will decide to improve the text and perhaps resubmit it.

Sincerely. 

Reviewer 3 Report

- Thank you for the opportunity to read this interesting paper
- I would suggest to rework the title
- Also a change to more precise keywords is suggested
- It is recommended to shorten the abstract and rework it following the guidelines of the journal
- Please add references (lines 46-51; 76-78)
- Can you bring the focus down to one research question (and maybe two sub-questions)?
- Please expand the introduction as to the guidelines of the paper and also restructure the introduction a bit (more paragraphs etc. for better readability)
- Could you also implement the idea of crowdsourcing for innovation in your literature review? f.ex. https://doi.org/10.3390/jrfm14020087
- The mentioned Jollychic (line 151) seems misplaced - if you want to keep it, please also use proper references
- please rework expression in line 167
- please strengthen the argument in paragraph beginning line 175
- please rework the paragraph beginning line 187 --> some part should go to the methodology, explain there, why you focus on the Chinese market
- line 298 - how did you select these companies?
- page 7 is empty
- line 317 - why is it Chinese-focussed, if Chinese companies are not even 2/3s of the sample?
- It is suggested to include a table of the measurement items as an overview
- please consider reworking the expression in chapter 5
- please expand chapters 5.1 (with more references to existing research) and 5.2
- please add as well some summary to answer the research question

Round 2

Reviewer 1 Report

First of all, I feel that this submission has improved the last version of the manuscript. However, some further considerations are the following:

On the one hand, in section 2. Literature review, the research proposal is described and presented, but it seems reasonable to move the model/conceptual framework to the end of this section, with a short introduction about the three hypotheses, before Section3. Research methods. On the other hand, in figure 3, it is also recommended to briefly describe the findings even for readers who are not familiar with the topic.

Reviewer 2 Report

Dear Authors, 

First of all, I would like to say that I am impressed with the revisions you have implemented in your study. Your paper looks as if it was a brand new one. Additionally, thank you for the answers to my opinions. I am pleased that you decided to incorporate all of my suggestions.

The resubmitted study is now much more coherent, supported by an exhaustive literature review with a clearly described research process. You have also developed the conclusions section, so now I can say that the paper is well-structured and contributes to academia.

I found two things that could be still improved: 

  • you could emphasize academic gaps even more visibly and then refer to them in the conclusions (also in a more noticeable way)
  • please translate the Chinese title into English (for example, to put them in brackets). As the journal's audience is international, I think it would be helpful for them to understand your work thoroughly.

Congratulations on your hard work!

Sincerely. 

Reviewer 3 Report

Dear Authors,

Thank you very much for your revision. I think, the paper is now good.

All the best

Author Response

Dear reviewer,

Thank you so much for encouraging our work.

Best regards.